# High Levels of Anti-SARS-CoV-2 Receptor-Binding Domain (RBD) Antibodies One Year Post Booster Vaccinations among Hospital Workers in Indonesia: Was the Second Booster Needed?

**DOI:** 10.3390/vaccines11081300

**Published:** 2023-07-30

**Authors:** Amila Hanifan Muslimah, Marita Restie Tiara, Hofiya Djauhari, Muhammad Hafizh Dewantara, Evan Susandi, Agnes Rengga Indrati, Bachti Alisjahbana, Arto Yuwono Soeroto, Rudi Wisaksana

**Affiliations:** 1Department of Internal Medicine, Hasan Sadikin General Hospital, Faculty of Medicine, Universitas Padjadjaran, Bandung 40161, Indonesia; 2Research Center for Care and Control of Infectious Disease, Universitas Padjadjaran, Bandung 40161, Indonesia; 3Department of Clinical Pathology, Hasan Sadikin General Hospital, Faculty of Medicine, Universitas Padjadjaran, Bandung 40161, Indonesia

**Keywords:** antibody, COVID-19 vaccination, healthcare worker

## Abstract

In August 2022, Indonesia prioritized healthcare workers to receive the second booster dose. We conducted a sequential serosurvey to understand the dynamics of the antibody titers. The first serosurvey, which was conducted in June 2021, 1–6 months after Sinovac vaccination, showed a median antibody level of 41.4 BAU/mL (interquartile range (IQR): 10–629.4 BAU/mL). The second serosurvey was conducted one month (August 2021) after the first Moderna booster vaccine and showed a median level of 4000 BAU/mL (IQR: 3081–4000 BAU/mL). The last serosurvey was conducted a year (August 2022) after the booster and showed a median level of 4000 BAU/mL (IQR: 4000–4000 BAU/mL). In this last survey, only 39 (11.9%) of healthcare workers had antibody levels below the maximum level of 4000 BAU/mL. Thus, one year after the first booster dose, we did not observe the waning of antibody levels. The average increase was perhaps because of natural infection. Based on these considerations, we believe that a second booster dose was not necessary for this category of subjects at that time. Because vaccine supply is often limited, priority could be given to the general population or other high-risk patient groups with low antibody titers based on serological tests.

## 1. Introduction

The severe acute respiratory syndrome coronavirus 2 (SARS-CoV-2) infection is an ongoing health problem worldwide, including in Indonesia [1]. The presence of anti-SARS-CoV-2 antibodies postinfection or because of vaccination has been confirmed to provide some protection against infection and against severe coronavirus 2019 (COVID-19) disease [2]. Vaccination is therefore considered necessary and important to prevent infection. However, current evidence is still lacking about how many boosters are necessary for healthcare workers (HCWs) and the community at large. This is challenging to determine because anti-SARS-CoV-2 antibody levels are affected by the type of vaccine received, exposure to natural infection, time since last vaccination/exposure, and internal factors, such as age and comorbidity [3,4].

HCWs were greatly affected during the first wave in 2020 and during the outbreak of the SARS-CoV-2 Delta strain in July–September 2021. Our HCWs in this survey received both doses of the Sinovac vaccine from January to June 2021. Sinovac was chosen as the vaccine for HCWs, because it was the most readily available vaccine in Indonesia at the time. Next, they received the Moderna booster in July 2021. During the Omicron outbreak in early 2022, many HCWs were also infected with COVID-19 but did not develop severe illness. In July 2022, the Indonesian Ministry of Health decided to prioritize a second mRNA vaccine for HCWs in Indonesia [5,6].

With the announced vaccine scheme and the periodic exposure to COVID-19 during outbreaks, we questioned the level of the anti-receptor-binding domain (RBD) titers among HCWs. Information about the level of anti-SARS-CoV-2-S-RBD antibodies can assist in evaluating the need for additional vaccination [7]. Herein, we aimed to describe the SARS-CoV-2-S-RBD antibody levels among HCWs in three consecutive serosurveys conducted at different time points that can provide information for considering the need for a second booster.

## 2. Materials and Methods

This research was conducted at Hasan Sadikin General Hospital, which is the top provincial referral hospital for general illnesses as well as for COVID-19 in Bandung, West Java, Indonesia.

### 2.1. Ethics Approval

This study was approved by the Health Research Ethics Committee of Hasan Sadikin General Hospital (number: LB.02.01/X.6.5/117/2022). 

### 2.2. Study Design

This study is a single-center, observational analysis study with three independent cross-sectional designs using primary data obtained by examining levels of SARS-CoV-2 IgG S-RBD antibodies in hospital HCWs. The data comprised basic participant information from interviews and results of the anti-SARS-CoV-2-S-RBD antibody levels. All participants were vaccinated using Sinovac in January–June 2021. This study used a convenience sampling method.

Our first serosurvey was conducted in July 2021, immediately before the third vaccine dose or the first booster with the Moderna vaccine. The second serosurvey was conducted in August 2021, one month after provision of the first Moderna booster vaccine. The third serosurvey was conducted in August 2022, 12 months after the first Moderna booster, immediately before the second Moderna booster that was scheduled by the Ministry of Health.

### 2.3. Inclusion Criteria

Eligible participants were HCWs of Hasan Sadikin General Hospital that were enrolled by voluntary participation. All participants were vaccinated with both doses of the inactivated virus vaccine (Sinovac) and one booster dose of mRNA-1273 (Moderna) full dose. All hospital staff who followed these vaccination schemes and were willing to participate in the study were included in the survey. There were no exclusion criteria for participation. Subjects with comorbidities were also included in this survey. 

### 2.4. Operational Definition

We used the following operational definition. Physicians were all medical doctors (MDs), mostly specialists but also included general practitioners working in the hospital. All other HCW participants were nonphysician hospital staff. The COVID-19 wards were areas that operated as isolation rooms assumed to have the highest risk of spreading COVID-19. These were the general ward, intensive care unit, and emergency isolation rooms specific for COVID-19 patients. History of close contact with confirmed cases included face-to-face contact with probable cases or confirmed cases within a 1 m radius and for 15 min or more, direct physical touch with probable or confirmed cases, or people who provided direct care to probable or confirmed cases without using standard personal protective equipment. Last COVID-19 infection (i.e., having COVID-19 illness in between surveys) was considered positive in patients who had a positive RT-PCR result or positive rapid antigen result with at least three COVID-19 symptoms based on the interview [8].

### 2.5. Antibody Level Examination

Determination of anti-SARS-CoV-2-S-RBD antibodies utilized a point-of-care quantitative immunochromatographic assay (FastBioRBD™) made by Wondfo, Guangzhou Biotech, China that was rebranded for distribution by PT Biofarma, Bandung, Indonesia (Persero). This equipment has a value range between 1 and 200 arbitrary units (AU), wherein 1 AU/mL is regarded as the minimal level for being seropositive. We later multiplied the result by 20 to obtain the standard WHO binding antibody unit (BAU) [9,10] Therefore, 20 BAU/mL was the lower limit for seropositivity, while the maximum detection limit was 4000 BAU/mL. A similar immunofluorescence assay test (Wondfo Finecare anti-RBD) was utilized and tested in Qatar [11]. To validate the FastBioRBD™ test, 71 randomly selected serum samples were tested using the GenScript cPass SARS-CoV-2 surrogate viral neutralization test (SVNT) Kit (Genscript Biotech, Leiden, The Netherlands). Briefly, serum samples as well as negative and positive controls were diluted 1:10 in a sample dilution buffer, mixed 1:1 with an HRP-RBD working solution, and incubated at 37 °C for 30 min. More information can be obtained in [12].

### 2.6. Data Analysis

The baseline characteristics of participants were factors that could influence the SARS-CoV-2-S-RBD antibody levels, namely age, sex, occupation, comorbidity, last COVID-19 infection, work zone, history of close contact, and levels of SARS-CoV-2 IgG S-RBD. Data on baseline characteristics were obtained based on information from the interview by using a validated questionnaire. The anti-SARS-CoV-2-S-RBD antibody levels are described in tables and figures. We analyzed baseline characteristics of participants and looked for risk factors of having a low (less than 4000 BAU/mL) antibody response 12 months after the third vaccination. Numerical data were tested for normality with the Kolmogorov–Smirnov test. Categorial data were analyzed using the chi-square and Fisher’s exact tests if unfulfilled. We used the Mann–Whitney test to conduct a bivariate analysis of numerical data. Variables with *p* < 0.25 in the bivariate analysis were further tested using multivariate logistic regression, with a significance value of *p* 0.05. We used SPSS software version 25 and Graphpad Prism software version 9.2.0 to conduct the analysis.

## 3. Results

For the first survey, we enrolled 570 persons on the first booster vaccination day. After blood collection, these subjects directly received the first booster of Moderna (full dose). One month after this first booster, we enrolled 355 persons in the second survey. One year after the first booster, we enrolled and collected serum samples from 330 persons in the third survey immediately before the second booster of Moderna (see Figure 1). One hundred and forty-eight people participated in all surveys. Participants’ demographic and clinical characteristics are summarized in Table 1.

The median age of the participants was 38 (IQR: 32–50) years followed by 40 (IQR: 33–52) and 41 (IQR: 34–52) in the first, second, and third surveys, respectively. There were more female than male participants with similar a proportion of physicians vs. nonphysicians in all three surveys. Nonphysician participants were mostly administrative staff, nurses, laboratory staff, and others (midwives, nutritionists, radiographers, and security staff) with a proportion of 22.1%, 16.6%, 3.0%, and 11.2%, respectively, in the third survey. Most of the participants worked in the non-COVID wards, while about a third worked in the COVID-19 isolation wards (see Table 1). The distribution of antibody levels based on the time of examination is presented in the Figure 1

In the first survey, the SARS-CoV-2-S-RBD antibody level median was 41.4 (range: 0.2–2.049.6) BAU/mL. One hundred and eight (32.7%) participants showed negative results for the SARS-CoV-2-S-RBD antibody (<20 BAU/mL), and 175 (30.7%) participants did not reach the median value (<41.4 BAU/mL). No participant reached the maximum detection level of 4000 BAU/m (see Figure 1). From the interview, we found 127 (22.2%) participants reported having a confirmed COVID-19 infection since the beginning of the pandemic (see Table 1).

In the second survey, we observed a median value of 4000 (range: 840.4–4000) BAU/mL of anti-SARS-CoV-2-S-RBD antibodies. One hundred and fifty (42.3%) participants had antibody levels below the maximum detection level of 4000 BAU/mL. No participant showed negative anti-SARS-CoV-2-S-RBD antibody results (see Figure). Within a one-month duration between the first and second surveys, 34 participants (9.5%) reported having a confirmed COVID-19 illness (see Table 1).

In the third survey, we obtained a median SARS-CoV-2-S-RBD antibody level of 4000 (range: 144.8–4000) BAU/mL, while only 39 (11.8%) of participants had antibody levels below the maximum detection level of 4000 BAU/mL (see Table 1). Similar to the second survey, we found no participant having negative anti-SARS-CoV-2-S-RBD antibody results (see Figure 1).

Within the duration of one year between the second and the third surveys, 124 (37.5%) participants reported having a confirmed COVID-19 illness. The fewer reported infections in the second survey was probably because of the shorter time interval between the first and second surveys than between the second and third surveys.

A bivariate analysis test was performed on factors potentially related to antibody levels in the last serosurvey. The bivariate analysis showed a trend that being a physician was associated with lower SARS-CoV-2-S-RBD antibody levels. There were more physicians who had antibody levels < 4000 BAU/mL) (15.4%) compared to non-physicians (8.5%; *p* value = 0.052) (Appendix A). However, the multivariate analysis did not confirm this as an independent factor (adjusted OR 0.512 (95% CI 0.258–1.015) (Appendix A).

Seventy-one randomly selected serum samples underwent validation tests with Genscript C-Pass. FastBioRBD™ compared to C-Pass showed a correlation with an r-value of 0.780 (95% confidence interval: 0.678–0.862) (Appendix A).

## 4. Discussion

We found high levels among HCWs in our hospital one year post booster vaccination. Many researchers and clinicians believe that the antibody levels stimulated by the COVID-19 vaccine may gradually decline over several months [3,4]. However, our observation shows that they were maintained or even slightly increased. 

Antibody levels were examined in participants about 6 months after being vaccinated with both Sinovac vaccines and were shown to reach a marginal level. A substantial proportion of participants still showed negative serology. This finding is consistent with the performance of the Sinovac and Coronavac vaccines. A study in Turkey conducted in March 2021 on the anti-RBD antibody levels 28 days post Sinovac vaccine showed that 99.4% had a seropositive result with a median of 154.78 BAU/mL, while only 32.4% of the participants reached maximum antibody levels [13]. A more similar study to ours conducted by Fonseca et al. in Brazil in September 2021 showed anti-S-RBD antibody levels 6 months after a complete dose of the Sinovac vaccine were modest, with a median of 66.7 BAU/mL, and 16 (2%) participants became seronegative [14]. Differences in these findings can also be affected by the time interval between vaccination to the antibody testing and the local COVID-19 transmission dynamics in each country/region [15].

The increase in antibody levels one month after receiving the first Moderna booster was expected. Over half of the respondents showed a maximum readable level of RBD antibodies in this survey. However, there were still a few subjects who showed only a moderate increase in antibody response. Adding a booster of mRNA vaccine in subjects who received the inactivated vaccine was shown to increase their antibody levels. A study in Chile showed that the group of subjects who received two Sinovac vaccines and a booster with the BNT162b2 RNA vaccine had higher antibody levels one month after the third vaccination than the group that received two Sinovac vaccines and the third booster with Sinovac [16]. Several studies have also shown the effect of the Moderna vaccine, which demonstrated a higher antibody response than the BNT vaccine [17,18].

The third survey was conducted in August 2022, a year after the last booster of the Moderna vaccine and after the Omicron outbreak in early 2022. In this survey, we found an even higher anti-RBD antibody level among hospital staff, and there were no signs of decreasing antibody levels. The increased antibody level between the second and third surveys was most probably due to continuous COVID-19 exposure. The high antibody level found in our study was similar to the national serosurvey conducted in Indonesia. The Indonesian serosurvey was conducted in October–November 2021 and July 2022. These surveys also revealed an increasing antibody level in general communities across Indonesia [19,20]. A similar study in Italy conducted from January to November 2021 also showed similar results where anti-SARS-CoV-2-RBD antibody levels increased months after receiving a booster vaccine [21].

Our study showed that participants who experienced a COVID-19 infection had higher anti-SARS-CoV-2-RBD antibody levels. This is especially shown in the first serosurvey after the Sinovac vaccine. This phenomenon is not new as many studies have shown this observation [22,23]. However, in our study, we can see that the effect is less prominent among subject who has been exposed to the mRNA vaccine. We hypothesize that the highest antibody level at the third serosurvey is a cumulation of vaccination and natural transmission, which was still highly prevalent in the year 2022. 

We did not find specific factors that increase the likelihood of having low (<4000 BAU) antibody levels. There is a trend that being a physician was related to lower antibody levels. This finding can be explained in that perhaps physicians had a more structured working scheme and fewer social gatherings. Several other reports have consistently shown that in developing countries, infections were mostly due to casual contact rather than the working environment [24]. 

Our study extensively used the quantitative immunochromatographic assay that was easy-to-use and affordable. The test uses a cassette-based dry reagent with a small portable reader that can be utilized in a point-of-care setting. Compared to other tests, this test has good validity. Our validation test of 71 samples with Genescript C-Pass showed a good correlation result. Previously, a validation test incorporating 150 subjects in Qatar showed a very good correlation with GenScript C-Pas SVNT as the reference test with an R-value of 0.70 (95% CI: 0.60–0.80) [11]. Because anti-SARS-CoV-2-RBD antibody levels can determine the level of protection [25], we can integrate serosurveys within a vaccination program to accurately evaluate its achievements and forecast the need for an additional booster.

Our study has some limitations. We enrolled the persons invited to the survey based on their willingness to be tested. Thus, the number of samples, especially for the second and third surveys, may not represent the whole hospital staff population. However, within our observation, we believe we represented the entire range of types of workers and exposure levels. An additional analysis we conducted on the 148 participants, who followed the serosurvey consistently, showed a similar pattern (Appendix A). Information about previous COVID-19 infections was collected through interviews or self-reporting. This method may not be the most accurate given that there might be information bias as self-reporting may lead to a lower prevalence than registered reports [19,20]. Lastly, our method of determining anti-SARS-CoV-2-RBD antibodies used a method, which has a maximum detection limit of 4000 BAU/mL; therefore, we could not measure antibody levels beyond this figure. This limitation may have decreased the analysis sensitivity of the factors related to lower antibody levels. However, it will not affect our definition of high antibody levels as we have learned that 2360 BAU already corresponds to the 90% COVID-19 protection level [25], while this level is already beyond this limit. 

## 5. Conclusions

Most HCWs did not show decreased antibody levels approximately one year after the booster. This was most likely because of the high transmission of SARS-CoV-2 and the omicron variant outbreak in early 2022. Most HCWs had very high levels of the SARS-CoV-2-S-RBD antibody 12 months after the third COVID-19 vaccine dose. Based on this result, we should reconsider whether a fourth vaccine dose was necessary for all. According to a recommendation from the Strategic Advisory Group of Experts (SAGE) World Health Organization (WHO) [26], additional boosters beyond the first booster are no longer routinely recommended for the medium priority-use group. For a more efficient use and appropriate distribution of the vaccine, taking due account of the epidemiology background and the evolution of the virus, we suggest incorporating antibody testing and only providing the booster vaccine to those with low levels of anti-RBD antibodies. 

## Figures and Tables

**Figure 1 vaccines-11-01300-f001:**
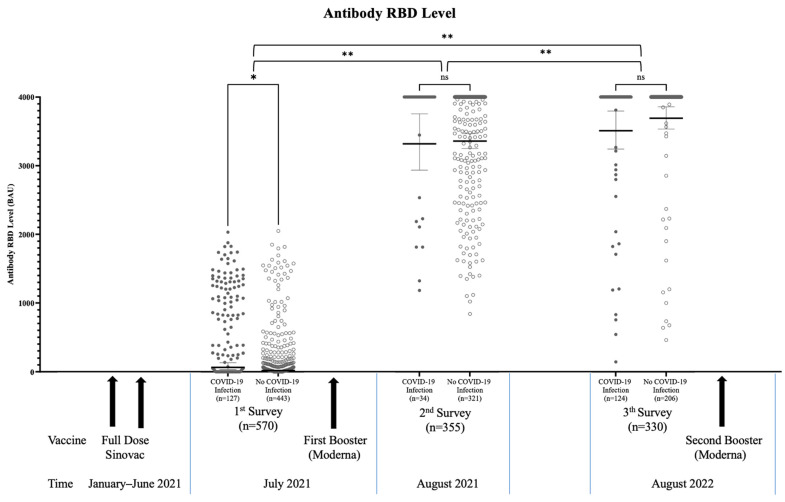
Anti-SARS-CoV-2-S-RBD Antibody Levels Across Serosurveys. Sequential serosurvey of SARS-CoV-2-S-RBD antibodies before the first Moderna booster, one month after the first Moderna booster, and 12 months after first Moderna booster. Most HCWs had a very high level of SARS-CoV-2-S-RBD antibodies 12 months after the third COVID-19 vaccine dose. The second Moderna booster was given right after the blood was sampled in the third survey. Note: Thin horizontal lines depict geometric mean, grey dots was participants with COVID-19 infection, and white dots was participants without COVID-19 infection, and * *p* value < 0.05, ** *p* value < 0.001.

**Table 1 vaccines-11-01300-t001:** Baseline characteristics of participants in the first, second, and third surveys.

Variables	First Survey (n = 570)	Second Survey (n = 355)	Third Survey (n = 330)
Sex			
Male, n (%)	215 (37.8)	120 (33.9)	112 (34.0)
Female, n (%)	355 (62.2)	235 (66.1)	218 (66.0)
Age category, median (IQR)			
18–30 years, n (%)	88 (15.5)	55 (15.5)	39 (12)
31–40 years, n (%)	223 (39.1)	130(36.7)	122 (36.9)
41–50 years, n (%)	118 (20.7)	72 (20.3)	70 (21.2)
51–60 years, n (%)	108 (18.9)	69 (19.4)	79 (23.9)
>60 years, n (%)	33 (5.8)	29 (8.1) **	20 (6.0) ^+^
Occupation			
Physician, n (%)	283 (49.6) *	208 (58.5) **	155 (46.9) ^
Nonphysician, n (%)	287 (50.4)	147 (41.5)	175 (53.1)
Work zone, n (%)			
In COVID-19 ward	147 (25.8) *	119 (33.5)	102 (30.9)
Not in COVID-19 ward	254 (44.5)	155 (43.7) **	134 (40.6) ^+^
Administrative work	169 (29.7) *	81 (22.8)	94 (28.5)
History of close contact with confirmed case, n (%)	210 (36.8) *	15 (4.2) **	68 (20) ^+^
Last COVID-19 infection (having previous COVID-19 infection in between surveys), n (%)	127 (22.2) *	34 (9.5) **	124 (37.5) ^+^
Antibody level (median, IQR) BAU/mL	41.4 * (10–629.4)	4000 (3081–4000)	4000 (4000–4000)
Antibody level classification			
≥4000 BAU/mL	0 (0)	205 (57.7) **	291 (88.1)
<4000 BAU/mL	570 (100)	150 (42.3)	39 (11.9)

Notes: n = frequency, % = percentage, ^ *p* value < 0.05 in multivariate analysis in the third survey, * *p* value < 0.05 between the 1st and 2nd surveys, ** *p* value < 0.05 between the 2nd and 3rd surveys, and ^+^
*p* value < 0.05 between the 3rd and 1st surveys.

## Data Availability

The data used to support the findings of this study are included in the article.

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
