# Peer review of "High Levels of Anti-SARS-CoV-2 Receptor-Binding Domain (RBD) Antibodies One Year Post Booster Vaccinations among Hospital Workers in Indonesia: Was the Second Booster Needed?"

_vaccines, 2023, doi:10.3390/vaccines11081300_

Round 1

Reviewer 1 Report (New Reviewer)

Thank you for asking me to review this article.

The study topic is very topical not only with regard to epidemiological aspects but also with regard to the protection of workers in the healthcare sector. Assessing the antibody response achieved over time to SARS-CoV-2 among healthcare workers, investigating the differences in age group, gender, and other risk factors as a professional category (doctors, nurses, and/or other hospital employees) is of paramount importance not only from a health surveillance perspective but also for the implementation and enhancement of the best prevention strategies.

However, the manuscript is not adequately designed to pursue the intended objective. The methodology has major shortcomings and the results are confusing.

The objective was indeed, as stated in the title, to evaluate the immune response of HCWs in terms of duration and protection in order to assess whether or not a second booster dose is needed. However, the authors chose to do this by selecting not cohorts of subjects but three independent samples. They also mix HCWs with patients.

There is evidence on the same aspects from more rigorous studies. Therefore, the work, given its limitations, is of more local than international interest.

We therefore suggest moving towards other journals after better clarifying and stating the study design and reporting the results consistently.

In the methods in fact they state that this is "a single-center, observational analysis study with a prospective design...", when in fact it is three independent cross-sectional studies, in view of the fact that a cohort of subjects was not followed. This is not explicitly stated by the authors, but is evident from the numbers represented in the table where in the row "Non Doctors" the number of observations increases from 147 in the second phase to 175 in the third

It is evident, with even the patients, who were not the target of the study, that the results are biased

Therefore, more revisions are needed before we even think of proceeding with a further review process for submission to a local journal.

Furthermore, in my opinion, the contents of the introductory section should be expanded; the hypotheses that the research intended to test should be expanded so as to make the context clearer. The epidemiological trend of SARS-CoV-2 cases in Indonesia for example is not described, nor is there any mention of the vaccination coverage of health workers in Indonesia, or at least in the hospital where the survey was conducted. In line 40 the authors refer generically to "...Our HCWs received both doses of Sinovac vaccine from January...". which cohort of HCWs are they referring to? To all those working in Indonesia or to the workers involved in the survey?

Moreover, on line 33 the authors indicate '...Vaccination in therefore considered mandatory', but in my opinion this concept should be expressed more cautiously and clearly. In fact, the assumption that vaccination is an important and indispensable life-saving weapon at the disposal of public health and the population in the fight against infectious diseases is undisputed, and the COVID-19 pandemic has shown how vaccination prevention has been essential in saving millions of lives worldwide. However, the concept of 'mandatory' in my opinion should be used with caution and prudence, especially in the current context where the imposition of vaccination is considered one of the main determinants of Vaccine Hesitancy in the general population. I suggest the authors rephrase this sentence to avoid misunderstandings and misinterpretation. Therefore this concept should not only be rephrased but also expanded upon with an adequate bibliography of references.

For another line 47-49 the authors ask whether a booster dose is necessary and, although the research question is legitimate and interesting, the sentence put in this way almost suggests that a high antibody titre does not necessarily imply a booster dose for vaccination. This concept could be misinterpreted and lead to a hesitation to undergo a full cycle as suggested by the vaccination schedule without having first performed antibody research. The authors should reformulate the contents presented in the introduction without neglecting aspects that deserve further investigation.

In my opinion, the methods section is incomplete. The authors describe a cohort of healthcare workers from Hasan Sadikin General Hospital in Bandung claiming that it is the best hospital. On what basis? What characteristics? It is suggested to add a paragraph on the study setting detailing the context and referring to both the catchment area of the centre where the study was conducted and the distribution of the survey population from which the study sample was drawn.

The methods also lack the paragraph on statistical analyses, which are instead presented in the results. Table 1 is unclear and the information is presented in a haphazard manner. I suggest the authors make the table intelligible.

Minor revisions:

Regarding form, the title is consistent with the content of the article, however it contains an error: 'Sars-Co-V2'.

Authors are recommended to make acronyms explicit at least the first time they are used in the text.

The tables are arranged in the text in a scattershot manner and make reading not very smooth. It is recommended that tables be included where they are cited, alternatively the authors may decide to include them in the supplementary material.

As far as the discussion is concerned, I suggest extending the empirical value of the results with reference to future perspectives as well.

Author Response

Point 1.

Thank you for asking me to review this article. 

The study topic is very topical not only with regard to epidemiological aspects but also with regard to the protection of workers in the healthcare sector. Assessing the antibody response achieved over time to SARS-CoV-2 among healthcare workers, investigating the differences in age group, gender, and other risk factors as a professional category (doctors, nurses, and/or other hospital employees) is of paramount importance not only from a health surveillance perspective but also for the implementation and enhancement of the best prevention strategies.

However, the manuscript is not adequately designed to pursue the intended objective. The methodology has major shortcomings and the results are confusing.

The objective was indeed, as stated in the title, to evaluate the immune response of HCWs in terms of duration and protection in order to assess whether or not a second booster dose is needed. However, the authors chose to do this by selecting not cohorts of subjects but three independent samples. They also mix HCWs with patients.

Response:

Thank you for your review. We feel the same way with your opinion that assessing the antibody response achieved over time is important in many aspects of vaccination program and prevention of disease. Despite our design limitation, we believe that our result can convey the message that the Anti-SARS-CoV2-antibody remain high among most HCW of Hasan Sadikin Hospital and this should become a consideration factor for deciding to upscale the second booster vaccination for all.

About the participant, thank you for pointing this out. We have made a wrong statement about the presence of some patient at the COVID-19 wards at the first paragraph of the result. Actually there was no patient participating in this study. We have corrected this statement in line 140-142. All participants were HCWs of Hasan Sadikin Hospital as mention in the method section in inclusion criteria.

Point 2.

There is evidence on the same aspects from more rigorous studies. Therefore, the work, given its limitations, is of more local than international interest. 

We therefore suggest moving towards other journals after better clarifying and stating the study design and reporting the results consistently.

In the methods in fact they state that this is "a single-centre, observational analysis study with a prospective design...", when in fact it is three independent cross-sectional studies, in view of the fact that a cohort of subjects was not followed. This is not explicitly stated by the authors, but is evident from the numbers represented in the table where in the row "Non Doctors" the number of observations increases from 147 in the second phase to 175 in the third

It is evident, with even the patients, who were not the target of the study, that the results are biased

Therefore, more revisions are needed before we even think of proceeding with a further review process for submission to a local journal.

Response:

Thank you for your review. We think that this short report can provide information from the real world situation where anti SARS-CoV-2-antibody level remain high for most HCW in the Hospital. In the discussion we attached a reference which has also show similar result where high anti-SARS-CoV-2-antibody were found increasingly high in general population in Indonesia that can be accessed internationally. We think this message is appropriate for general international audience, firstly; to describe the rarely seen phenomenon we have learnt from COVID-19 pandemic and secondly; to emphasise the need for serosurvey as an evaluation tool for vaccination program and further adjustment of vaccination targets. We have corrected the method section in manuscript stating that this was a “three independent cross-sectional studies”.

Point 3

Furthermore, in my opinion, the contents of the introductory section should be expanded; the hypotheses that the research intended to test should be expanded so as to make the context clearer. The epidemiological trend of SARS-CoV-2 cases in Indonesia for example is not described, nor is there any mention of the vaccination coverage of health workers in Indonesia, or at least in the hospital where the survey was conducted. In line 40 the authors refer generically to "...Our HCWs received both doses of Sinovac vaccine from January...". which cohort of HCWs are they referring to? To all those working in Indonesia or to the workers involved in the survey?

Response:

Thank you for your concern. We have describe the research question and aim of the study in the last paragraph of the introduction section. More information about SARS-CoV-2 in Indonesia are emphasise in discussion section. We have referred our thoughts to several references listed in the discussion section. Information in line 40 refer to HCWs that working in this survey. This information also describe in figure.

Point 4

Moreover, on line 33 the authors indicate '...Vaccination in therefore considered mandatory', but in my opinion this concept should be expressed more cautiously and clearly. In fact, the assumption that vaccination is an important and indispensable life-saving weapon at the disposal of public health and the population in the fight against infectious diseases is undisputed, and the COVID-19 pandemic has shown how vaccination prevention has been essential in saving millions of lives worldwide. However, the concept of 'mandatory' in my opinion should be used with caution and prudence, especially in the current context where the imposition of vaccination is considered one of the main determinants of Vaccine Hesitancy in the general population. I suggest the authors rephrase this sentence to avoid misunderstandings and misinterpretation. Therefore this concept should not only be rephrased but also expanded upon with an adequate bibliography of references. 

Response:

Thank you for your concern, we have corrected in line 33 to “Vaccination is therefore considered necessary and important to prevent infection.”

Point 5

For another line 47-49 the authors ask whether a booster dose is necessary and, although the research question is legitimate and interesting, the sentence put in this way almost suggests that a high antibody titre does not necessarily imply a booster dose for vaccination. This concept could be misinterpreted and lead to a hesitation to undergo a full cycle as suggested by the vaccination schedule without having first performed antibody research. The authors should reformulate the contents presented in the introduction without neglecting aspects that deserve further investigation.

     Response:

Thank you for your concern, in this study is aimed to understand the dynamics of the antibody titres and in real world situation. This information may provide knowledge for considering the need for a second booster. We have corrected in the introduction section.

Point 6

In my opinion, the methods section is incomplete. The authors describe a cohort of healthcare workers from Hasan Sadikin General Hospital in Bandung claiming that it is the best hospital. On what basis? What characteristics? It is suggested to add a paragraph on the study setting detailing the context and referring to both the catchment area of the centre where the study was conducted and the distribution of the survey population from which the study sample was drawn. 

Response:

Thank you for your correction, we have corrected statement “the best hospital” in the materials and methods section in manuscript, “This research was conducted at Hasan Sadikin General Hospital, which is the top provincial referral hospital in West Java province “.

Point 7

The methods also lack the paragraph on statistical analyses, which are instead presented in the results. Table 1 is unclear and the information is presented in a haphazard manner. I suggest the authors make the table intelligible.

Response:

Thank you for your correction, we have corrected in material section. We add information in data analysis.

“We analysed baseline characteristics of participants and looks for risk factors of having low (less than 4000 BAU/mL) antibody response 12 months after The 3rd Vaccination. Numerical data has been tested for normality with the Kolmogorov Smirnov test. Categorial data was analysed using bivariate with unpaired categorical comparative bivariate, Chi square test, if unfulfilled using the Exact-Fisher test. Numerical data was analysed with the Mann-Whitney test. Bivariate analysis with p < 0.25 were analysed using multivariate logistic regression, with a significance value of p 0.05. Statistics were performed using SPSS software”

Minor revisions:

  1. Regarding form, the title is consistent with the content of the article, however it contains an error: 'Sars-Co-V2'. 

Response:

Thank you for your correction, we have corrected SARS CoV-2 in the title.

  1. Authors are recommended to make acronyms explicit at least the first time they are used in the text.

Response:

Thank you for your correction, we have added the complete name in the first appearance for acronyms which were used later in the text.

  1. The tables are arranged in the text in a scattershot manner and make reading not very smooth. It is recommended that tables be included where they are cited, alternatively the authors may decide to include them in the supplementary material.

Response:

Thank you for your concern, we have corrected the tables position and alignment.

  1. As far as the discussion is concerned, I suggest extending the empirical value of the results with reference to future perspectives as well.

Response :

Thank you for your input, We have added additional statement in line 266 as such “Since anti SARS-CoV2-RBD antibody levels can determine the level of protection (Shuo Feng, 2021), we can integrate serosurveys within a vaccination program to accurately evaluate its achievements and  forecast the need of an additional booster”

Reviewer 2 Report (Previous Reviewer 2)

The authors addressed all of my queries.

Author Response

Thank you very much for your evaluation.

Reviewer 3 Report (New Reviewer)

Lines 55-57: I don't think that the authors should state that Hassan Sadikin General Hospital is the best hospital in Bandung unless they can provide a reference that it is the best hospital.

How many total HCW are employed at Hassan Sadikin Hospital and are the participates in this study representative of the total number of HCW with respect to age, sex and occupation?

What statistical tests were used in figure 1 when comparing the infected versus non-infected at each survey time point?  For surveys 2 and 3, the figure shows that there was no significance between the infected and non-infected groups.  However, because a large portion of the values are at or above the upper limit of 4000 BAU/mL, the statistical test used might not be valid. Can the serum samples be diluted and the antibody tests be performed again to determine the true value of the antibody results? 

There were 570 participants in the first survey, 355 in the second and 330 in the third.  Were the 330 participants in the third survey also in the first and second surveys?  If not, then the authors should state how many individuals participated in all 3 surveys.

Line 208 states "participants with high antibody levels experienced more COVID-19 infections"  Can you authors please show this data and the statistical analysis to back up this statement?

In line 235-237, the authors state that "Most HCWs did not show decreased antibody levels approximately one year after booster.  This was most likely because of the high transmission of SARS-CoV-2 and the omicron variant outbreak in early 2022."  I believe that this statement is unsubstantiated because in survey 3, 62.5% of the participants did not report having a COVID-19 infection and yet these participants had high antibody levels.  Were nucleocapsid antibody tests performed on the participants to see if they might have had asymptomatic infections?

Author Response

  1. Lines 55-57: I don't think that the authors should state that Hassan Sadikin General Hospital is the best hospital in Bandung unless they can provide a reference that it is the best hospital.

Response:

Thank you for your concern, we have corrected in the materials and methods section in manuscript.

  1. How many total HCW are employed at Hassan Sadikin Hospital and are the participates in this study representative of the total number of HCW with respect to age, sex and occupation?

Response:

There are 3.297 employed at Hasan Sadikin Hospital. This study is represent to age with IQR (32-50) in first survey, (33-52) in second survey, and (34-52) in third survey. For sex, this study is also represent female and male, with female have higher proportion this study is included doctor, nurse, administrator worker, laboratory staff, midwife, radiographer, nutritionist, and security, but the proportion is not same with total employed proportion’s because this study is used consecutive sampling.

  1. What statistical tests were used in figure 1 when comparing the infected versus non-infected at each survey time point?  For surveys 2 and 3, the figure shows that there was no significance between the infected and non-infected groups.  However, because a large portion of the values are at or above the upper limit of 4000 BAU/mL, the statistical test used might not be valid. Can the serum samples be diluted and the antibody tests be performed again to determine the true value of the antibody results? 

Response:

Mann-Whitney test were used to comparing the infected versus non infected at each survey time point. Our equipment to test antibody level has a value range between 20 BAU/mL and 4000 BAU/mL. We think that it was not unfeasible and unnecessary to conduct diluted sample testing. Firstly, we use a Fluorescent Immunoassay Meter which has not been validated to do a diluted sample testing to get accurate anti SARS-CoV-2 antibody level. Secondly; we have learnt that > 2360 BAU do correspond to a 90% protection level (Shuo Feng, Nature Medicine, https://doi.org/10.1038/s41591-021-01540-1). Reading of 4000 BAU is already beyond that level therefore we can rightly categorize our participant as having high antibody level. 

Reference :

Feng, S., Phillips, D.J., White, T, Sayal H, Aley PK, Bibi S, et al. Correlates of protection against symptomatic and asymptomatic SARS- CoV-2 infection. Nat Med. 2021;27: 2032–2040.

  1. There were 570 participants in the first survey, 355 in the second and 330 in the third.  Were the 330 participants in the third survey also in the first and second surveys?  If not, then the authors should state how many individuals participated in all 3 surveys.

Response:

Yes, 330 participant in third survey were also in the first or second survey. There were 148 individuals that participated in all 3 surveys. We have made a graph using this 148 sample as figure supplement figure B2. The graph showed similar pattern. This results show that in the first survey (before the third vaccination) it was shown that antibody levels were slightly increased in those infected with COVID-19 compared to those who were not infected with COVID-19, but in the second survey (1 month after the third vaccine) and third survey (12 months after third vaccine), there was no significant difference in antibody levels between those infected with COVID-19 and those not infected with COVID-19.

Figure B2. SARS-Co-V-2-S-RBD Antibody Level Across Serosurveys among 148 HCWs

  1. Line 208 states "participants with high antibody levels experienced more COVID-19 infections"  Can you authors please show this data and the statistical analysis to back up this statement?

Response:

Thank you for your concern, We would like to ask you to see the supplementary table A2. In this table we showed that there were more participant having COVID-10 infection (106, 85.4%) among group with anti SARS-CoV2-RBD antibody level ≥4000 BAU/mL. Whereas there were 18 (14.6%) participants with history of COVID019 infection in the group with anti SARS-CoV2-RBD antibody level <4000 BAU/mL.

  1. In line 235-237, the authors state that "Most HCWs did not show decreased antibody levels approximately one year after booster.  This was most likely because of the high transmission of SARS-CoV-2 and the omicron variant outbreak in early 2022."  I believe that this statement is unsubstantiated because in survey 3, 62.5% of the participants did not report having a COVID-19 infection and yet these participants had high antibody levels.  Were nucleocapsid antibody tests performed on the participants to see if they might have had asymptomatic infections?

Response

Thank you for your concern, the third survey is done after omicron variant outbreak. Data of COVID 19 infection is done by self-reporting of SARS CoV-2 PCR test on people who are willing to be examined. So there is a possibility that people who are asymptomatic but have positive PCR results were not reported in this study. One of the reason might be that the high of transmission of SARS-CoV-2 and the omicron variant outbreak in early 2022. In this study, nucleocapsid test cannot be performed, because the tools are not available.

Round 2

Reviewer 1 Report (New Reviewer)

The authors recognised the methodological problems of the paper and partially remedied them by providing an adequate explanation.

However, there remains a serious problem with the current validity of the paper itself, as the authors have set it up. In fact, given the retrospective nature of the study, taking into account also the fact that the epidemiological picture has changed since the research conditions, and taking into account also the fact that the virus is evolving with new variants, the suggestion of not administering a second booster dose to that cohort of healthcare workers today appears pleonastic: not least because they have in fact already got it.

The authors must therefore amend their conclusions in the abstract and in the text to provide possible valuable suggestions for the future.:

In the Abstract

instead of

“Based on this fact, we suggest reconsidering the need for the second booster dose. Given that the vaccine supply is limited, it can be better utilized for the general public or other high-risk groups of patients who have a low level of antibody titers based on serological testing.”

Replace with:

“Based on these considerations, we believe that a second booster dose was not necessary for this category of subjects at that time. Since vaccine supply is often limited, priority could be given to the general population or other high-risk patient groups with low antibody titres on the basis of serological tests.”

In the text:

Row 53: provide information on whether a booster is necessary  àwas

Row 238-241: Based on this result, we should re-consider whether a fourth vaccine dose is necessary for all. For efficient use and distribution of the vaccine, we suggest to incorporate antibody testing and only providing booster vaccines to those who have low anti-RBD antibody level

Based on this result, we should re-consider whether a fourth vaccine dose was necessary for all. According to reccomendation from SAGE (https://www.who.int/publications/i/item/who-wer9822-239-256), additional boosters beyond the first booster are no longer routinely recommended for the medium priority-use group. For a more efficient use and appropriate distribution of the vaccine, taking due account of the epidemiological background and the evolution of the virus, we suggest incorporating antibody testing and only providing the booster vaccine to those with low levels of anti-RBD antibodies.

Author Response

The authors recognised the methodological problems of the paper and partially remedied them by providing an adequate explanation.

However, there remains a serious problem with the current validity of the paper itself, as the authors have set it up. In fact, given the retrospective nature of the study, taking into account also the fact that the epidemiological picture has changed since the research conditions, and taking into account also the fact that the virus is evolving with new variants, the suggestion of not administering a second booster dose to that cohort of healthcare workers today appears pleonastic: not least because they have in fact already got it.

The authors must therefore amend their conclusions in the abstract and in the text to provide possible valuable suggestions for the future.:

  1. In the Abstract

instead of

“Based on this fact, we suggest reconsidering the need for the second booster dose. Given that the vaccine supply is limited, it can be better utilized for the general public or other high-risk groups of patients who have a low level of antibody titers based on serological testing.”

Replace with:

“Based on these considerations, we believe that a second booster dose was not necessary for this category of subjects at that time. Since vaccine supply is often limited, priority could be given to the general population or other high-risk patient groups with low antibody titres on the basis of serological tests.”

Response :

Thank you for your input. Your suggestion correctly put the study result in the context of the current situation. we have corrected the statement in the abstract.

  1. In the text:

Row 53: provide information on whether a booster is necessary   

Response :

Thank you for your concern, we have corrected the statement at the end of the introduction section. Now it is written as “Herein, we aimed to describe the SARS-CoV-2-S-RBD antibody level among HCWs in three consecutive serosurveys conducted at different time points that can provide information for considering the need for a second booster”

  1. In the text

Row 238-241: Based on this result, we should re-consider whether a fourth vaccine dose is necessary for all. For efficient use and distribution of the vaccine, we suggest to incorporate antibody testing and only providing booster vaccines to those who have low anti-RBD antibody level

Based on this result, we should re-consider whether a fourth vaccine dose was necessary for all. According to reccomendation from SAGE (https://www.who.int/publications/i/item/who-wer9822-239-256), additional boosters beyond the first booster are no longer routinely recommended for the medium priority-use group. For a more efficient use and appropriate distribution of the vaccine, taking due account of the epidemiological background and the evolution of the virus, we suggest incorporating antibody testing and only providing the booster vaccine to those with low levels of anti-RBD antibodies.

Response :

Thank you for your concern, we have corrected the statement in the conclusion. Now it is written as below; “Based on this result, we should reconsider whether a fourth vaccine dose was necessary for all. According to a recommendation from Strategic Advisory Group of Ex-perts (SAGE) World Health Organization (WHO) [28], additional boosters beyond the first booster are no longer routinely recommended for the medium priority-use group. For a more efficient use and appropriate distribution of the vaccine, taking due account of the epidemiology background and the evolution of the virus, we suggest incorporating antibody testing and only providing the booster vaccine to those with low levels of an-ti-RBD antibodies”.

  1. Additional Correction

Besides the suggested correction, we did some additional correction in the text and the note of the figure.

Round 3

Reviewer 1 Report (New Reviewer)

The authors have responded to all comments raised and, in my opinion, the quality of manuscript has improved. I believe that the paper can be published in its current form.

This manuscript is a resubmission of an earlier submission. The following is a list of the peer review reports and author responses from that submission.

Round 1

Reviewer 1 Report

This study was conducted to determine the level of SARS CoV2 S-RBD antibody reached by the first vaccine, after the first booster, and before the second booster to understand the dynamics of the antibody level. They found that antibody levels in three serosurvey were 41.4 BAU/mL, 4000 BAU/mL, and 4000 BAU/mL, respectively. This study provided useful data evidence for evaluating whether the second booster dose is really necessary. As a brief communication, I think this study is very interesting and valuable. However, some concerns should be addressed.

1.        This study had a fatal flaw in its experimental design: individuals with a history of close contact with confirm COVID-19 case and last COVID 19 infections were included. It has been showed that infection with SARS-CoV-2 can induce a significantly higher antibodies levels in the coming months, which may interfere the real immune response induced by the COVID-19 vaccine. Therefore, the reliability of the data derived based on this study is not high, and the sample size of this study is too small. This deficiency cannot be addressed by modifying the article or supplementing the data, and I have serious concerns about publishing this study.

2.        A timeline for vaccination and serologic testing must be included so that readers can understand the study context more clearly.

3.        First serosurvey was conducted in July 2021, the second serosurvey was conducted in August 2021, and the third serosurvey was conducted in August 2022. Why choose these time points to detect the level of antibodies? Authors must give sufficient evidence to support this performance in the main text.

4.        Although authors claimed that “Eligible participants were health workers of Hasan Sadikin General hospital collected by voluntary participation.” in lines 72-73, but detailed Inclusion and exclusion criteria of individuals was not included in this study. It has been reported that patients with other diseases such as HIV, Immunodeficiency, Autoimmune diseases, may have an impact on the immune response to vaccines.

5.        “Healthcare workers” in line 41, but “health care workers” in line 46. “healthcare workers” should be shorten for “HCWs” in the second or more times appear in the text.

6.        In Abstract section, author claimed that “Healthcare workers in Indonesia acquired a complete 2 doses of Sinovac in early 2021 and first booster dose of Moderna in July 2021.”, but in the Introduction section, it was showed that “Next, they acquire Moderna Booster in August 2021.” (Line 45). Please confirm the real time for booster of Moderna vaccine. I suggest that the authors check the time and relevant data of the full text to avoid inconsistencies.

7.        The section of “Materials and Methods” should be divided into several parts, such as Ethics and Participants, Inclusion and exclusion criteria, Measurement of serum antibody levels, Statistical Analysis, etc.

8.        The writing of this article is too colloquial, and English is very poor. Therefore, the English language must be improved by an English native speaker. For example, “we came to wonder how high is our anti RBD-Level? And, is this second booster really necessary?” in lines 50-51.

9.        In Table 1, what is “y.o”? Is there any significant difference between two subgroups or among three or more groups? I suggest authors to perform a statistical analysis. For example, t test or rank sum test can be used for measurement data, and chi-square test or Fisher's exact test can be used for count data. Based on these differences with P < 0.05, the further analysis must explore the antibodies levels between subgroups, such as age groups, Occupation, and Work-zone.

10.      Figure 1, P value and SD or SEM should be showed. Furthermore, the statistical methods should be added in the figure legend.

 The writing of this article is too colloquial, and English is very poor. Therefore, the English language must be improved by an English native speaker.

Reviewer 2 Report

The authors assessed antibody levels for the healthcare workers in a hospital. The results might be useful in the field, however, several variables were not analyzed. I have some comments/queries to the current version of the manuscript.

- lines 41-48: The authors can elaborate more for the vaccine scheme in Indonesia provided by the Ministry of Health. I suggest including the following information: Two types of vaccines were available: (1) inactivated virus technology platform (Sinovac) was available on XX (date), (2) mRNA technology platform (Moderna) was available on YY (date). Move line 74 to here, then, it is not necessary to introduce Sinovac and Moderna in subsequent text.

- lines 93-94: I do not understand the rationales of x20 of AU to fit for the WHO BAU. Any reference(s) for this manipulation?

- lines 97-98: To be clear, suggest to revise as: To validate the FastBioRBDtm test, 71 randomly selected samples were tested using both the GenScript cPass SARS-CoV-2 Neutralization Antibody Detection 98 Kit (Genscript Biotech, Leiden, Netherlands) and the FastBioRBDtm test. To correspond the evaluation results mentioned in lines 160-162, the authors should list out the expected r- value in the method section.

- the results/conclusion might be correlate with the objectives, however, there were some concerns for the results shown in the 3rd survey.

(1) the number of dots shown in the 3rd survey (N=330) were much lower than the 2nd survey (N=355)

(2) varying antibody levels were shown between 4000 and 300, it means that some of them had diminished level, the authors made an incorrect statement in line 197.

(3) SARS-CoV-2 infections demonstrated highest in survey 3, readers were not able to visualize if the antibody level were due to normal drop or infection, suggest to use different color of dots to indicate no infection (N=206) and had infection (N=124). The authors should also indicate this information to 1st and 2nd surveys.

(4) when comparing between the 1st and 3rd surveys, the SARS-CoV-2 infection rates were similar, 22.2% and 37.5%, the authors should discuss the reasons for the difference of antibody levels between them, i.e. why 3rd survey is higher than 1st survey? due to the mRNA vaccine? due to different SARS-CoV-2 strains (i.e. delta vs omicron)?

N/A